# Evaluation of the Diagnostic Performances of the SD-Bioline^®^HBeAg Rapid Test Used Routinely for the Management of HBV-Infected Individuals in Burkina Faso

**DOI:** 10.3390/diagnostics13193144

**Published:** 2023-10-07

**Authors:** Abdoulaye Dera, Armel M. Sanou, Mathuola N. G. Ouattara, Abdoul K. Ilboudo, David B. Lankoande, Dieudonné Ilboudo, Delphine Napon-Zongo, Michel K. Gomgnimbou

**Affiliations:** 1Laboratoire de Recherche sur les Maladies Infectieuses et Parasitaire (LR-MIP), Institut de Recherche en Science de la Santé, Bobo-Dioulasso 2779, Burkina Faso; ablodera@gmail.com (A.D.); ouattaranina98@gmail.com (M.N.G.O.); 2Département des Laboratoires, Centre “Assaut-Hépatites”, Bobo-Dioulasso 2285, Burkina Faso; 3Laboratoire de Recherche sur les Maladies Infectieuses et Parasitaire (LR-MIP), Institut de Recherche en Science de la Santé, Ouagadougou 7192, Burkina Faso; ilboudokader@yahoo.fr; 4Département Méthodologie et Gestion des Données, Centre “Assaut-Hépatites”, Bobo-Dioulasso 2285, Burkina Faso; 5Service des Urgences Médicales, Centre Hospitalier Universitaire de Bogodogo, Ouagadougou 314, Burkina Faso; bakanoulankoande@gmail.com; 6Département Clinique, Centre “Assaut-Hépatites”, Bobo-Dioulasso 2285, Burkina Faso; ddilboudo@gmail.com (D.I.); zpdelphine@yahoo.fr (D.N.-Z.); 7District Sanitaire de Banfora, Direction Régionale de la Santé des Cascades, Banfora 117, Burkina Faso; 8Institut Supérieur des Sciences de la Santé (INSSA), Université Nazi Boni, Bobo-Dioulasso 1091, Burkina Faso; gomikir@yahoo.fr; 9Laboratoire de Biologie Moléculaire, Centre Muraz, Bobo-Dioulasso 2054, Burkina Faso

**Keywords:** hepatitis B, HBeAg, rapid diagnostic test, ELFA, Burkina Faso

## Abstract

Hepatitis B e antigen (HBeAg) is a marker of wild-type hepatitis B virus replication. In resource-limited countries where access to enzyme-linked immunosorbent assay (ELISA) remains a challenge, rapid diagnostic tests (RDT) constitute a good alternative. The HBeAg status is employed to evaluate eligibility for antiviral therapy and to prevent the transmission of hepatitis B from mother to child (PMTCT). The objective of this study was to assess the diagnostic performance of the SD-Bioline^®^HBeAg RDT commonly used for detecting HBeAg in laboratories in Burkina Faso. The sample panel used was collected from HBsAg-positive patients received in the laboratory for the detection of HBeAg with the rapid test. The samples were retested for HBeAg using the VIDAS HBe/Anti-HBe enzyme-linked fluorescent assay (ELFA) (Gold standard). Then, the viral load (VL) of HBV DNA was determined using the GENERIC HBV CHARGE VIRLAE kit (GHBV-CV). The diagnostic performances of the SD-Bioline^®^HBeAg and its agreement with the gold standard were calculated with their 95% confidence intervals. Overall, 340 sera obtained from HBsAg-positive patients were included in this evaluation Compared to the VIDAS HBe/Anti-HBe ELFA test, the sensitivity (Se) and specificity (Sp) of the SD-Bioline^®^HBeAg test were 33.3% and 97.9%, respectively. The concordance between the two tests was 0.42. Depending on the viral load, the Se and Sp varied from 8.8% and 98.3% for a VL < 2000 IU/mL to 35.5% and 98.4% for a VL > 2,000,000 IU/mL. The results showed a low sensibility of the SD-Bioline^®^HBeAg RDT test, indicating that its use is inappropriate for the clinical management of HBV-infected patients. They also highlight the urgent need to develop HBeAg rapid tests with better sensitivities.

## 1. Introduction

Hepatitis B virus (HBV) infection remains a public health problem for many low- and middle-income countries (LMIC). According to the World Health Organization, 296 million people will be living with chronic hepatitis B in 2019, i.e., 3.8% of the world’s population [1,2]. Despite the availability of an effective HBV vaccine, the number of new infections per year was estimated at 1.5 million, and most new infections globally occur in Sub-Saharan Africa (SSA) (990,000) [1,2]. Africa and the West Pacific are the regions most affected by this infection. In Africa, around 82 million people are chronically infected with HBV [1]. This infection is associated with the risk of hepatic decompensation, cirrhosis, and hepatocellular carcinoma (HCC) [3]. It is also the leading cause of cirrhosis and hepatocellular carcinoma (HCC) in Sub-Saharan Africa [4].

The World Health Assembly has adopted the Global Health Sector Strategy for the Elimination of Viral Hepatitis by 2030. As part of this strategy, it is planned to reduce the number of new cases of hepatitis B by 95% and the number of deaths by 65% by 2030 [5]. Achieving this goal will require an ambitious increase in screening and treatment activities for HBV infection [6]. Ideally, these chronic carriers should be identified and medical interventions implemented to avoid the risk of progression to complications [7]. In the case of chronic HBV infection, viral DNA quantification is the reference test for assessing the stage of infection, and also for treatment decision [1]. However, access to molecular tests for DNA quantification remains problematic in resource-limited countries. HBeAg, which is a marker of wild-type virus replication and essential for the classification of HBV infection [6,8], has been proposed in combination with alanine aminotransferase (ALT) assay for therapeutic decision-making in resource-limited countries [1,9]. For HBeAg detection, immunological tests such as rapid diagnostic tests (RDT), enzyme-linked immunosorbent assay (ELISA), and enzyme-linked fluorescent assay (ELFA) are used. Due to their low cost and ease of use, RDTs are widely used in resource-limited countries. Although offering advantages, the diagnostic performances of RDTs can be influenced by factors such as low detection limits and genetic diversity [10,11]. Therefore, it is necessary to evaluate these RDTs with local biological samples before their widespread use. Previous studies evaluating HBeAg RDTs have been conducted in the African setting. In Senegal, diagnostic performance evaluation of three RDTs revealed poor sensitivities ranging from 29.8–42.5% [9]. In Malawi, similar results were reported for three other HBeAg RDTs (28.0–72.0%) [6].

In Burkina Faso, HBV infection is endemic with an estimated prevalence of 9.1% in the general population [12,13]. The Ministry of Health (MoH) adopted a strategic plan to combat viral hepatitis in Burkina Faso in 2017. The strategic axes of this plan focused on screening and diagnosis, prevention, and standards and protocols for the management of viral hepatitis at different levels of care. At community level and in level 1 and 2 health facilities, screening and diagnosis are based mainly on the use of RDTs. In level 3 health facilities, screening and diagnosis are based on RDTs, enzyme linked-immunosorbent assay (ELISA), and polymerase chain reaction (PCR) for HBV DNA quantification [14].

Despite the widespread use of the SD-Bioline^®^HBeAg RDT for the clinical management of HBV-infected patients in Burkina Faso, no studies evaluating the test’s diagnostic performance under real-life conditions have been carried out. Thus, this study was carried out to evaluate the diagnostic performance of the SD-Bioline^®^HBeAg RDT currently used in laboratories in Burkina Faso.

## 2. Methods

### 2.1. Study Design

This was an evaluation study of a diagnostic tool that was carried out from January 2020 to October 2022. The sample panel used for this evaluation was obtained from HBsAg-positive patients received at the biomedical analysis laboratory of the “Assaut-Hépatites” Center with a duly completed examination form for the detection of HBeAg with the SD-Bioline^®^HBeAg rapid test. The “Assaut-Hépatites” center is a specialist center for the clinical management of viral hepatitis. HBsAg+ patients are regularly referred to the center’s laboratory for biological analyses. For the purposes of the study, patients received for HBeAg testing were included progressively until the desired sample size was reached. Approximately 8 mL of whole blood from each consenting patient was collected by venipuncture from the elbow. The blood sample was then centrifuged at 4000 rpm for 5 min and the serum obtained was aliquoted in 3 cryotubes. One was used directly for the SD-Bioline^®^HBeAg test according to the manufacturer’s instructions and the second was used for the detection of HBeAg using enzyme-linked fluorescent assay (ELFA) (Gold standard) and the quantification of HBV viral DNA by real-time PCR. Sociodemographic information was collected using a structured questionnaire by the laboratory staff.

### 2.2. SD-Bioline^®^HBeAg RDT

The SD-Bioline^®^HBeAg (Standard Diagnostics, Inc., Suwon, Republic of Korea) rapid diagnostic test is a one-step in vitro immunochromatographic test designed for the detection of HBeAg in human serum or plasma. It is a single-step, easy-to-use test that can be stored at 2 °C to 30 °C. The test was performed according to the manufacturer’s instructions and the Good Laboratory Practices (GLP). After removing the test device from the foil pouch, 100 µL of the collected sample was added to the sample well, and the result was interpreted after 5–20 min. Interpretation of the test results was based on the appearance of lines visible to the naked eye. A test was negative if the control zone line appeared in the absence of the test zone line. A positive test was characterized by the appearance of two lines, one in the test area and one in the control area. The test was invalid if the control zone line was absent. The performance characteristics of the test provided by the manufacturer were as follows: sensitivity (Se) = 95.5% (95% CI: 88.9–98.2%); specificity (Sp) = 98.6% (95% CI: 96.6–99.5%).

### 2.3. Gold Standard: VIDAS HBe/Anti-HBe (ELFA)

All samples were retested for HBeAg using the VIDAS HBe/Anti-HBe kit. VIDAS HBe/Anti-HBe is a qualitative automated test on VIDAS family instruments (BioMérieux SA, Marcy-l’Étoile, France), for the detection of HBeAg or anti-HBe antibodies in human serum or plasma by Enzyme Linked Fluorescent Assay (ELFA). To perform the test, the MINI VIDAS was first calibrated using the standards provided in the kit: S1 (HBeAg), C1 (positive control), C2 (negative control). Then, 150 µL of vortexed serum was added to each sample well of the corresponding cartridge and the analysis was started. Results were obtained after 90 min. Two fluorescence measurements in the reading cuvette are performed for each assay. The first one takes into account the background due to the substrate cuvette before the substrate is brought into contact with the cone (SPR). The second reading is taken after the substrate is incubated with the enzyme present in the cone (SPR). The calculation of the relative fluorescence value (RFV) is the result of the difference of the two measurements. The index of the test is calculated by dividing the RFV of the sample or control by the RFV of the standard: index (i) = RFV of the sample/RFV of the standard S1. The result is negative if i < 0.1 and positive if i ≥ 0.1. The analytical sensitivity of the HBeAg assay, determined according to the PEI standard provided by the manufacturer, was 0.25 PEI U/mL, and its specificity was greater than 98% (100%, 98.1%, and 99.5%). All tests were performed according to the manufacturer’s instructions.

### 2.4. Quantification of HBV Viral DNA

Viral DNA extraction was performed on the GenoXtract^®^ automated instrument using GXT NA extraction kits (Hain Lifescience, Nerhen, Germany). During extraction, an internal control supplied by the manufacturer was associated with each sample to validate the extraction and amplification process. To detect and quantify HBV DNA, quantitative real-time polymerase chain reaction (qPCR) was performed with GENERIC HBV CHARGE VIRALE (GHBV-CV) kit (BIOCENTRIC, Bandol, France) using FluoroCycler^®^XT (Hain Lifescience, Nerhen, Germany) platform in a reaction volume of 20 µL. The detection limit of the technique is 95 IU/mL (1.98 log_10_ UI/mL). The GHBV-CV assay uses an oligonucleotide detection hydrolysis probe labelled 5′ with a reporter fluorophore and 3′ with a quencher group. During the combined hybridization phase of PCR, the probe is cleaved by the 5′ to 3′ exonuclease activity of Taq DNA polymerase, which releases and separates the reporter from the quencher. This cleavage results in the emission of a detectable fluorescence which is proportional to the quantity of accumulated PCR products. The fluorescent signal is only detected if the amplified target sequence is complementary to the probe. For each PCR plate, a set of four standards is used to construct the standard curve (Ct as a function of concentration in log IU/mL). In this way, the viral load of each sample is obtained by extrapolating, from the standard curve, the concentration corresponding to the Ct value of each sample. Because of the genetic variability of HBV, the GHBV-CV test targets a well-defined sequence of the S gene. All steps were performed according to the manufacturer’s instructions and the GLP.

### 2.5. Statistical Analysis

Data were collected using Excel and statistical analyses were performed with R software version 4.0.1 (R Foundation for Statistical Computing, Vienna, Austria). We had used the R package “dplyr” to make mutations with variables. The comparison of the results of the SD-Bioline^®^HBeAg RDT with those of the ELFA allowed the calculation of the sensitivity, specificity, positive predictive value (PPV), and negative predictive value (NPV) with their 95% confidence intervals (CI) using the R package “questionr”. Agreement between the two tests was determined using Cohen’s Kappa test with the R package “vcd”. The Landis and Koch criteria stipulate the following: Kappa < 0, no agreement; 0 < kappa ≤ 0.2; slight agreement; 0.2 < kappa < 0.4, fair agreement; 0.4 < kappa < 0.6, moderate agreement; 0.6 < kappa < 0.8, substantial agreement; 0.8 < kappa < 1, near perfect agreement.

These performance characteristics were also determined based on different viral DNA quantity thresholds. The significance level of the analyses was set at *p* ≤ 0.05.

### 2.6. Ethical Approval

This study was approved by the institutional ethics committee of the “Institut de Recherche en Sciences de la Santé” (IRSS) (A01-2020/CEIRES du 23 January 2020). The participant consent was obtained for the use of the collected samples for this research.

## 3. Results

A total of 340 sera obtained from HBsAg-positive patients were included in this evaluation. The mean age of our participants was 33.75 ± 11.98 years, with extremes ranging from 7 to 81 years. Overall, 168 participants were male, with a sex ratio of 0.97. HBeAg detection frequencies were 4.70% (16/340) with SD-Bioline^®^HBeAg RDT and 10.58% (36/340) with VIDAS HBe/Anti-HBe (Table 1).

Compared to the VIDAS HBe/Anti-HBe ELFA test, the Se of the SD-Bioline^®^HBeAg RDT was 33.3% (95% CI: 18.5–50.9), the Sp was 98.6% (95% CI: 96.6–99.6), the PPV was 75.0% (47.6–92.7), and the NPV was 92.5% (89.1–95.1). The kappa value of agreement with reference test was 0.42 (95 CI: 0.25–0.59) (Table 2).

The diagnostic performance of the SD-Bioline^®^HBeAg RDT was also calculated according to different viral load thresholds. HBV DNA was detectable in 280 (82.3%) samples and undetectable in 60 (17.6%) (Table 3). The Se of the rapid test increased as the viral load increased, ranging from 8.8% for a VL ≥ 3.30 log_10_ IU/mL to 35.5% a VL ≥ 6.30 log_10_ IU/mL (Table 4, Figure 1). In contrast, Sp remained stable (approximately 98%) for all viral load thresholds (Table 4, Figure 1).

In addition, when comparing the VIDAS HBe/Anti-HBe ELFA test results with VL, we noted that 55.92% of VIDAS test results were negative when VL < 3.30 log_10_ IU/mL, 30.3% when VL [3.3–4.3 log_10_ IU/mL] and 13.8% for VL > 4.3 log_10_ IU/mL.

## 4. Discussion

This study evaluated the diagnostic performance of the SD-Bioline^®^HBeAg RDT compared to VIDAS HBe/Anti-HBe ELFA test as gold standard. To our knowledge, this is the first study of its kind in Burkina Faso. We found that the SD-Bioline^®^HBeAg RDT had a low sensitivity (33.3%) and a high specificity (98.6%). The sensitivity does not reach those provided by the manufacturer (95.5%), indicating that this RDT should be used with caution for the clinical management of HBV-infected patients. Similar sensitivities of 28.0% and 29.8% for SD-Bioline^®^HBeAg RDT have been reported by studies conducted in Malawi [6] and Senegal [9], respectively. Low sensitivity of HBeAg RDTs appears to be a widespread problem. Indeed, evaluations of the diagnostic performance of other RDTs used for HBeAg detection other than SD-Bioline have reported low sensitivities. These include the HBeAg Rapid Test (Biopanda Reagents Ltd., Belfast, UK) and the HBeAg serum rapid test (Creative Diagnostics, Shirley, NY, USA) for which sensitivities of 53.2% and 72.3% were observed, respectively [6]. The same is true for the Insight HBeAg (Tulip Diagnostics Ltd., Goa, India) and OneStep HbeAg (AMSUK Ltd., Antrim, UK) tests for which sensitivities of 31.1% and 42.5% were found [9], respectively. Furthermore, when comparing the performance of SD-Bioline RDT in relation to the geographical origin of samples, the highest sensitivity (47.1% vs. 29.8%) was observed on Asian samples. For OneStep RDT, the highest sensitivity was observed on samples from Senegal (42.5% vs. 35.3%) [9]. This shows that the genotypes of viruses circulating in different regions have an impact on RDT performance. The development of HBeAg RDTs should take into account samples of genotypes A and E, which are predominant in Sub-Saharan Africa.

The coefficient kappa observed in our study was 0.42. According to the criteria of Landis and Koch [15], the agreement between the SD-Bioline^®^HBeAg RDT and the VIDAS HBe/Anti-HBe ELFA test was moderate.

Moreover, the diagnostic performance of the SD-Bioline^®^HBeAg RDT was also calculated according to different viral load thresholds. We noted an increase in the sensitivity as the viral load increased, ranging from 8.8% for a VL ≥ 3.30 log_10_ IU/mL to 35.5% a VL ≥ 6.30 log_10_ IU/mL. Similar findings were reported by a Cambodian study where they noted an increase in sensitivities from 76.5% to 89.3% for VL > 5.30 log_10_ IU/mL and > 7.30 log_10_ IU/mL, respectively [16]. Although the sensitivity of RDT increased with the amount of HBV DNA detected in our study, 16.3% of HBeAg-negative sera had HBV DNA levels ≥ 4.30 log10 IU/mL (VL ≥ 20,000 IU/mL). This observation highlights the low analytical sensitivity of HBeAg RDT, probably due to mutations in the pre-core (Pre-C) and basal core promoter (BCP) regions, reflecting its poor ability to be correlated with HBV replication. This low sensitivity could therefore prevent patients with chronic hepatitis B who are eligible for treatment from accessing it. The main consequence would be to prevent the infection from progressing to the complications of cirrhosis and cancer.

The HBV genome is a partially circularized DNA, composed of four overlapping and shifting reading frames (ORFs): Pol (P), Core (C), Surface (S), and X. These ORFs encode the seven HBV proteins and also code for four promoter regions that initiate transcription and two enhancers that promote gene transcription [16]. Given its genomic organization, any mutation occurring in a specific region of the genome can potentially have an impact on other regions, thus affecting the viral cycle. The ORF C codes for two functionally different proteins: a particulate protein (Ag HBc) forming the nucleocapsid, and a soluble e-protein (Ag HBe) which is detected in the serum of patients infected with wild-type virus, during active viral replication [16,17]. Molecular analyses have revealed that mutations are frequently found in the Pre-C and BCP regions of the ORF C and are systematically associated with loss or reduced synthesis of HbeAg, despite the existence of active HBV replication [18,19]. Indeed, pre-C/BCP mutations are most commonly found in HbeAg-negative patients with detectable viral loads [20]. Among these mutations, we have the G1896A substitution (pre-C mutation), which leads to the appearance of a premature stop codon and the arrest of HbeAg translation, without impact on viral replication [20]. For a wild-type virus, the absence of HbeAg in the serum of an HBV-infected individual would correlate with a lack of viral replication. However, we found that 13.8% of HbeAg-negative sera with VIDAS Hbe/Anti-Hbe had a VL ≥ 4.30 log10 IU/mL. This observation is particularly common in people infected with mutant viruses unable to synthesize Hbe antigen [21]. Our results could therefore be explained by the presence of pre-C/BCP mutations in the HBV genome that impact HbeAg production [22]. The VIDAS HBe/Anti-Hbe kit was used to assess the risk of mother-to-child transmission of HBV in Ethiopia, and its sensitivity was found to be 50% for DNA levels above 5.30 log10 IU/mL (VL ≥ 200,000 UI/mL) [23,24]. Given the sensitivity of the VIDAS Hbe/Anti-Hbe kit, we could therefore say that the reduced sensitivity of the Bioline^®^HbeAg SD RDT evaluated in this—and previous—studies could probably be due to the variability of the strains used to develop these tests. Indeed, the VIDAS HBe/anti-HBe test is designed to detect HBeAg or anti-HBe antibody in the serum or plasma of people infected with wild-type virus. We could therefore say that mutations in ORF C could make it difficult to develop a well-preserved consensus sequence for standardizing HBeAg tests. We had not looked for HBV mutants in our HBeAg samples, but in view of previous studies [16,20], there may be pre-C/BCP mutants that would prevent HBeAg detection in our samples, especially those with a viral load greater than 4.30 log10 IU/mL.

By 2030, the WHO’s goal is to be able to diagnose over 90% of people with HBV and treat over 80% of those diagnosed and eligible for treatment [1,25]. The problem posed by pre-C and BCP mutations would call into question the algorithm (HBeAg/ALAT score) [1,9] for identifying people to be put on antiviral treatment proposed by the WHO, particularly in countries with a high prevalence of HBV. The literature suggests that BCP mutants (A1762T/G1764A) increase viral replication [21,26]. Although the clinical implication of pre-core and BCP mutants remains to be elucidated, several studies on the subject tend to link these mutants with the risk of developing cirrhosis or HCC [20,27,28]. Thus, the use of HBeAg as a marker of viral replication could aggravate the burden of HBV in resource-limited countries, particularly in its use for the prevention of mother-to-child transmission of HBV. For instance, in Burkina Faso, HBeAg-negative patients should be closely monitored, as around 77% rapidly progress to cirrhosis and HCC [29]. In addition, the advent of the COVID-19 pandemic has enabled several urban and rural laboratories to be equipped with molecular biology platforms. For better management of chronic hepatitis B, the health authorities should take advantage of these tools.

Our study has some limitations. The reference test used (VIDAS HBe/Anti-HBe ELFA) did not allow the quantification of HBeAg. This could allow a better appreciation of the analytical sensitivity of the RDT by determining the detection threshold. Furthermore, the sequencing of the Pre-Core region was not performed. This would give an idea of the prevalence of mutations, the type of mutations, and their impact on the performance of the diagnostic tests.

## 5. Conclusions

Without simple, affordable and reliable diagnostic tools to assess HBV viral replication, it is unlikely that the WHO’s global elimination targets can be met, especially as HBV infection represents a high burden on healthcare systems in resource-limited countries. In these contexts, RDTs can play an important role in intensifying the clinical management of HBV infection. The results showed a low sensitivity of the SD-Bioline^®^HBeAg test, indicating that its use is inappropriate for the clinical management of HBV-infected patients. These results and other evaluation studies highlight the urgent need to develop HBeAg rapid tests with better sensitivities. This would facilitate access to quality diagnostic tools for low-income populations and, consequently, better control of HBV infection. In the meantime, the use of point-of-care tests for HBV DNA quantification and/or the sharing of viral load platforms from HIV/TB programs could contribute to better patient management.

## Figures and Tables

**Figure 1 diagnostics-13-03144-f001:**
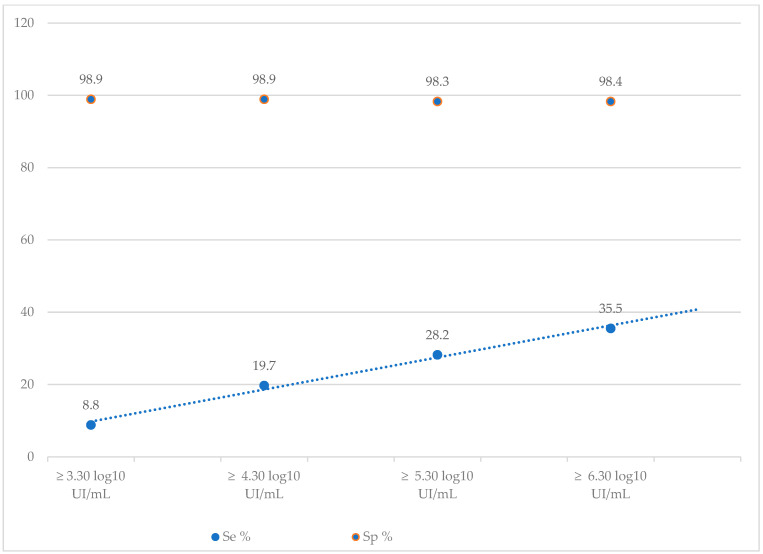
Evolution of the Se and Sp of the SD-Bioline^®^HBeAg RDT according to the viral load.

**Table 1 diagnostics-13-03144-t001:** Cross-tabulation of the SD-Bioline^®^HBeAg RDT results with the reference test results.

	Results	VIDAS HBe/Anti-HBe		
SD-Bioline^®^HBeAg RDT		Positive (%)	Negative (%)	Total
	Positive	12	4	16
	Negative	24	300	324
	Total	36	304	340

**Table 2 diagnostics-13-03144-t002:** Diagnostic performances of the SD-Bioline^®^HBeAg RDT compared to the reference test.

Performance Parameters	SD-Bioline^®^HBeAg RDT	
	Estimate (%)	95% CI
Se	33.3	18.5–50.9
Sp	98.6	96.6–99.6
VPP	75.0	47.6–92.7
VPN	92.5	89.1–95.1
Kappa	0.42	0.25–0.59

**Table 3 diagnostics-13-03144-t003:** Cross-tabulation of the SD-Bioline^®^HBeAg RDT results with the viral load of HBV.

Viral Load		SD-Bioline^®^HbeAg RDT		
		Positive	Negative	Total
VL ≥ 3.3 log_10_ UI/mL (VL ≥ 2000 UI/mL)	≥3.3 log_10_ UI/mL	14	145	159
	<3.3 log_10_ UI/mL	2	179	181
	Total	16	324	340
VL ≥ 4.3 log_10_ UI/mL (VL ≥ 20,000 UI/mL)	≥4.3 log_10_ UI/mL	13	53	66
	<4.3 log_10_ UI/mL	3	271	274
	Total	16	324	340
VL ≥ 5.3 log_10_ UI/mL (VL ≥ 200,000 UI/mL)	≥5.3 log_10_ UI/mL	11	28	39
	<5.3 log_10_ UI/mL	5	296	301
	Total	16	324	340
VL ≥ 6.3 log_10_ UI/mL (VL ≥ 2,000,000 UI/mL)	≥6.3 log_10_ UI/mL	11	20	31
	<6.3 log_10_ UI/mL	5	304	309
	Total	16	324	340

**Table 4 diagnostics-13-03144-t004:** Diagnostic performances of the SD-Bioline^®^HBeAg RDT compared to viral load of HBV.

Viral Load	SD-Bioline^®^HBeAg RDT							
	Se		Sp		VPP *		VPN **	
	Estimate (%)	95% CI	Estimate (%)	95% CI	Estimate (%)	95% CI	Estimate (%)	95% CI
VL ≥ 3.3 log_10_ UI/mL (VL ≥ 2000 UI/mL)	8.8	4.9–14.3	98.9	96.1–99.9	87.5	61.6–98.4	55.2	49.6–60.7
VL ≥ 4.3 log_10_ UI/mL (VL ≥ 20,000 UI/mL)	19.7	10.9–31.3	98.9	96.7–99.7	81.3	54.3–95.9	83.6	79.1–87.7
VL ≥ 5.3 log_10_ UI/mL (VL ≥ 200,000 UI/mL)	28.2	15.0–44.9	98.3	96.2–99.4	68.8	41.3–89.0	91.4	87.7–94.2
VL ≥ 6.3 log_10_ UI/mL (VL ≥ 2,000,000 UI/mL)	35.5	19.2–54.6	98.4	96.3–99.5	68.8	41.3–89.0	93.8	90.2–95.6

* Positive predictive value; ** Negative predictive value.

## Data Availability

Data generated during this study are available from the corresponding author on reasonable request.

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
