# Peer review of "Evaluation of the Diagnostic Performances of the SD-Bioline®HBeAg Rapid Test Used Routinely for the Management of HBV-Infected Individuals in Burkina Faso"

_diagnostics, 2023, doi:10.3390/diagnostics13193144_

Round 1
Reviewer 1 Report
1. This article reports on the performance of a RDT for HBeAg in Burkina Faso and is an important contribution to the literature.
2. I would add that most new infections globally occur in SSA (990,000 in WHO 2021 report).
3. "immunological tests such as RDT, ELISA and ELFA are used." please define these acronyms on first use.
4. The authors state that the Maskill method was used for analysis. I am not famililar with this method and cannot find a reference for this. Can the authors supply a reference or explain how this was calculated. The authors stated that R was used, were any packages used for this analysis?
5. The results section states "HBV DNA was detectable in 280 (8.2%) samples 180 and undetectable in 60 (17.6%)." I think there is a mistake in the 8.2% figure.
6. In table 3 and 4 the comparison has been made between the RDT and the HBV DNA as a reference test but it would be more meaningful to compare the RDT and ELFA at given HBV DNA thresholds. The abstract as currently written "Depending on the viral load, the Se and Sp varied from" gives the impression that this latter analysis has been done.
Secondarily, a separate analysis can be done comparing the HBeAg RDT and different clinical thresholds for HBV DNA eg. at 2000 and 20,000 and 200,000 IU/ml.
In table 3 the term "detectable" and "undetectable" has been used to refer to viral load thresholds. It should instead refer to above and below the given threshold.
In table 4 the terms "VPP" and "VPN" have been used without any explanation. Please use a footnote to describe the acronyms.
7. The authors report that a sample panel has been used. Can you provide more details on how these patients were recruited and selected. Was it consecutive, random or a convenience sample? How were they diagnosed and referred? This question is important for understanding the representativeness of the sample.
8. I do not think the analysis presented in figure 2 is informative. Please see point 6 above.
9. The first paragraph in teh discussion focuses on the prevalence of HBeAg which is already widely understood to be low in this setting and not directly focusing on the main study findings. I would instead remove the first paragraph and focus on a discussion of the findings relating to the RDTs. I would remove the sentence beginning "Indeed, the Landis and Koch criteria" and instead provide a reference.
10. The discussion focuses a lot on the relationship between the HBeAg and HBV DNA discussing pre-core mutants but this is a bit off-topic, since the main issue is the lack of sensitivity of HBeAg RDTs. I would instead focus more on potential reasons for low sensitivity and comment on the need for HBeAG RDTs adapted to the WHO African region (using genotype A and E samples) given the challenges of HBeAg detection using CLIA or ELISA due to limited access to laboratory infrastructure especially in rural areas.
11. "especially as HBV burden is a luxury in resource-limited countries." This sentence does not make sense, please rewrite.
12. For readability I would not abbreviate sensitivity "The results showed a low Se"
13. The authors having stated that the HBeAg RDT evaluated is clinically inappropriate due to low sensitivity should propose alternatives: either use HBV DNA with POC assays, share viral load platforms with HIV programmes, or use more liberal criteria for treatment eg. based on age and ALT alone.
There are a few minor corrections required, see comments above.
Author Response
- This article reports on the performance of a RDT for HBeAg in Burkina Faso and is an important contribution to the literature.
The authors sincerely thank reviewer #1 for this encouraging comment.
- I would add that most new infections globally occur in SSA (990,000 in WHO 2021 report).
We thank the reviewer #1 for this pertinent suggestion. We have taken it into account in the introduction section of line 52 to 53.
- "immunological tests such as RDT, ELISA and ELFA are used." please define these acronyms on first use.
These acronyms were first defined as suggested in the introduction to lines 70 to 71.
- The authors state that the Maskill method was used for analysis. I am not famililar with this method and cannot find a reference for this. Can the authors supply a reference or explain how this was calculated. The authors stated that R was used, were any packages used for this analysis?
The authors thank the reviewer for pointing out the lack of precision. To address this, corrections have been made in the statistical analysis section from line 169 to 174.
- The results section states "HBV DNA was detectable in 280 (8.2%) samples 180 and undetectable in 60 (17.6%)." I think there is a mistake in the 8.2% figure.
The mistake has been corrected and now we can read 280 (82.3%) at line 201.
- In table 3 and 4 the comparison has been made between the RDT and the HBV DNA as a reference test but it would be more meaningful to compare the RDT and ELFA at given HBV DNA thresholds. The abstract as currently written "Depending on the viral load, the Se and Sp varied from" gives the impression that this latter analysis has been done.
The authors would like to thank the reviewer for this comment, which we fully understand. However, we made the comparison between HBeAg RDT and viral DNA because the focus of the study is on the performance of the RDT.
Secondarily, a separate analysis can be done comparing the HBeAg RDT and different clinical thresholds for HBV DNA eg. at 2000 and 20,000 and 200,000 IU/ml.
The authors would like to thank the reviewer for his comments. If we had understood the comment correctly, we believe that these are the results presented in Table 4.
In table 3 the term "detectable" and "undetectable" has been used to refer to viral load thresholds. It should instead refer to above and below the given threshold.
We thank the reviewer for this pertinent suggestion, which has been taken into account in Table 3.
In table 4 the terms "VPP" and "VPN" have been used without any explanation. Please use a footnote to describe the acronyms.
A footnote has been used to describe the acronyms.
- The authors report that a sample panel has been used. Can you provide more details on how these patients were recruited and selected. Was it consecutive, random or a convenience sample? How were they diagnosed and referred? This question is important for understanding the representativeness of the sample.
We acknowledge Reviewer #1 for this relevant remark. More details have been provided for a better understanding of line 102 to 105.
- I do not think the analysis presented in figure 2 is informative. Please see point 6 above.
As suggested the figure 2 has been deleted.
- The first paragraph in teh discussion focuses on the prevalence of HBeAg which is already widely understood to be low in this setting and not directly focusing on the main study findings. I would instead remove the first paragraph and focus on a discussion of the findings relating to the RDTs. I would remove the sentence beginning "Indeed, the Landis and Koch criteria" and instead provide a reference.
We thank reviewer #1 for these suggestions and agree with him. As suggested, the first paragraph has been deleted. In addition, we have provided the reference and the Landis and Koch criteria have been moved to the section on statistical analysis.
- The discussion focuses a lot on the relationship between the HBeAg and HBV DNA discussing pre-core mutants but this is a bit off-topic, since the main issue is the lack of sensitivity of HBeAg RDTs. I would instead focus more on potential reasons for low sensitivity and comment on the need for HBeAG RDTs adapted to the WHO African region (using genotype A and E samples) given the challenges of HBeAg detection using CLIA or ELISA due to limited access to laboratory infrastructure especially in rural areas.
Thank you very much for your comment. We have tried to incorporate an additional explanation for the low sensitivity of HBeAg RDTs based on the impact of genotypes. In addition, pre-C mutants have been discussed to show their impact on test sensitivity.
- "especially as HBV burden is a luxury in resource-limited countries." This sentence does not make sense, please rewrite.
The authors are grateful to the reviewer #1 for this helpful comment. As suggested, the sentence has been rewrite as follow : “Without simple, affordable and reliable diagnostic tools to assess HBV viral replication, it is unlikely that the WHO's global elimination targets can be met, especially as HBV infection represents a high burden on healthcare systems in resource-limited countries”
- For readability I would not abbreviate sensitivity "The results showed a low Se"
We acknowledge Reviewer #1 for this relevant remark. As suggested, Se was then replaced by sensitivity at line 343.
- The authors having stated that the HBeAg RDT evaluated is clinically inappropriate due to low sensitivity should propose alternatives: either use HBV DNA with POC assays, share viral load platforms with HIV programmes, or use more liberal criteria for treatment eg. based on age and ALT alone.
To take account of this pertinent suggestion, a sentence has been added to the conclusion section, from line 348 to 350. It reads as follows: “In the meantime, the use of point-of-care tests for HBV DNA quantification and/or the sharing of viral load platforms from HIV/TB programs could contribute to better patient management”.

Reviewer 2 Report
The researchers have reported regarding their study entitled “Evaluation of the diagnostic performances of the SD-Bioline®HBeAg rapid test used routinely for the management of HBV-infected individ- 3 uals in Burkina Faso”
In the study the researchers evaluated the diagnostic performance of the SD-Bioline® HBeAg RDT compared to VIDAS HBe/Anti-HBe ELFA test as gold standard. The authors reported that the Bio- line® HBeAg SD RDT had a low sensitivity (33.3%) and a high specificity (98.6%)
This is an interesting study, however the results of the study are of importance only in many countries, particular the low income countries, which use the BiolineHBeAg rapid test.
As mentioned in the discussion, there are several studied published before and showed similar results, with low sensitivity.
The findings of the study are important, particularly for clinician who use this rapid test, however novelty is limited in the study.
The manuscript is well written and presented with clear logic and fluent structure. The data are presented appropriately.
There are several minor corrections that need to be addressed.
1. Line 180, Please correct the percentage – 82.2% not 8.2%
2. Line 223, the definition of Landis and Kock criteria definition is inappropriate to be included in the discussion, instead could be included in the methods section.
Indeed, the Landis and Koch criteria stipulate that: Kappa < 0, no agreement; 0 < kappa £ 0.2; slight agreement; 0.2 < kappa < 0.4, fair agreement; 0.4 < kappa < 0.6, moderate agreement; 0.6 < kappa < 0.8, substantial agreement; 0.8 < kappa < 1, near perfect agreement.
3. The conclusions, is too long, not all conclusions could be concluded from the results and the following sentence is fully unrelated the conclusions, should remove or included in the discussion “ To our knowledge, this is the first study to evaluate the diagnostic performance of an RDT for the detection 298 of HBeAg in Burkina Faso
Author Response
The researchers have reported regarding their study entitled “Evaluation of the diagnostic performances of the SD-Bioline®HBeAg rapid test used routinely for the management of HBV-infected individuals in Burkina Faso”
In the study the researchers evaluated the diagnostic performance of the SD-Bioline® HBeAg RDT compared to VIDAS HBe/Anti-HBe ELFA test as gold standard. The authors reported that the SD-Bioline® HBeAg RDT had a low sensitivity (33.3%) and a high specificity (98.6%)
This is an interesting study; however, the results of the study are of importance only in many countries, particular the low income countries, which use the SD- Bioline HBeAg rapid test.
As mentioned in the discussion, there are several studied published before and showed similar results, with low sensitivity.
The findings of the study are important, particularly for clinician who use this rapid test, however novelty is limited in the study.
The authors sincerely thank reviewer # 2 for these encouraging comments.
The manuscript is well written and presented with clear logic and fluent structure. The data are presented appropriately.
There are several minor corrections that need to be addressed.
- Line 180, Please correct the percentage – 82.2% not 8.2%
The percentage has been corrected.
- Line 223, the definition of Landis and Kock criteria definition is inappropriate to be included in the discussion, instead could be included in the methods section.
Indeed, the Landis and Koch criteria stipulate that: Kappa < 0, no agreement; 0 < kappa £ 0.2; slight agreement; 0.2 < kappa < 0.4, fair agreement; 0.4 < kappa < 0.6, moderate agreement; 0.6 < kappa < 0.8, substantial agreement; 0.8 < kappa < 1, near perfect agreement.
We thank the reviewer #1 for this pertinent suggestion. Indeed, the part has been moved in the statistical analysis section.
- The conclusions, is too long, not all conclusions could be concluded from the results and the following sentence is fully unrelated the conclusions, should remove or included in the discussion “To our knowledge, this is the first study to evaluate the diagnostic performance of an RDT for the detection 298 of HBeAg in Burkina Faso
As suggested, the sentence has been removed and included in the discussion.
